# Estimating the Spatial Distribution and Future Conservation Requirements of the Spotted Seal in the North Pacific

**DOI:** 10.3390/ani13203260

**Published:** 2023-10-19

**Authors:** Leyu Yang, Hongfei Zhuang, Shenghao Liu, Bailin Cong, Wenhao Huang, Tingting Li, Kaiyu Liu, Linlin Zhao

**Affiliations:** 1School of Advanced Manufacturing, Fuzhou University, Jinjiang 362200, China; yangleyu123456789@163.com (L.Y.); liukaiyu@fio.org.cn (K.L.); 2Key Laboratory of Marine Eco-Environmental Science and Technology, First Institute of Oceanography, Ministry of Natural Resources, Qingdao 266061, China; zhuanghf@fio.org.cn (H.Z.); shliu@fio.org.cn (S.L.); biolin@fio.org.cn (B.C.); hwh@fio.org.cn (W.H.); ytlitingting@163.com (T.L.); 3College of Environmental Science and Engineering, Ocean University of China, Qingdao 266100, China

**Keywords:** spotted seal, climate change, ecological niche divergence, species distribution models, conservation gap analysis

## Abstract

**Simple Summary:**

To understand the impact of climate change on marine mammals, we focused on the spotted seal population in the North Pacific. This ice-breeding species exhibits distinct variations across different regions. Our study aimed to quantify their ecological niches and conduct a conservation gap analysis. We found clear niche divergence among three populations and observed habitat contraction driven by climate change, potentially leading to breeding habitat loss in certain areas. Unfortunately, existing marine protected areas do not adequately cover most spotted seal habitats. By incorporating local adaptation into species distribution modeling, our research provides valuable insights for designing effective conservation policies to protect the different geographical populations of spotted seals in the face of climate change. This study highlights the importance of considering local adaptation in conservation and management strategies for marine mammal species.

**Abstract:**

Local adaptation has been increasingly involved in the designation of species conservation strategies to response to climate change. Marine mammals, as apex predators, are climatechange sensitive, and their spatial distribution and conservation requirements are critically significant for designing protection strategies. In this study, we focused on an ice-breeding marine mammal, the spotted seal (*Phoca largha*), which exhibits distinct morphological and genetic variations across its range. Our objectives were to quantify the ecological niches of three spotted seal populations, construct the species-level model and population-level models that represent different regions in the Bering population (BDPS), Okhotsk population (ODPS) and southern population (SDPS), and conduct a conservation gap analysis. Our findings unequivocally demonstrated a clear niche divergence among the three populations. We predicted habitat contraction for the BDPS and ODPS driven by climate change; in particular, the spotted seals inhabiting Liaodong Bay may face breeding habitat loss. However, most spotted seal habitats are not represented in existing marine protected areas. Drawing upon these outcomes, we propose appropriate conservation policies to effectively protect the habitat of the different geographical populations of spotted seals. Our research addresses the importance of incorporating local adaptation into species distribution modeling to inform conservation and management strategies.

## 1. Introduction

Climate change poses a significant peril to global biodiversity in the 21st century, unleashing a series of profound and unpredictable changes on our planet [1]. In comparison to terrestrial communities, marine communities stand as more vulnerable sentinels, acutely attuned to the environmental changes wrought by this climatic change [2]. The effects of climate change on the marine environment have changed the life history and distribution landscape of marine species, with the potential to cause habitat destruction and even species extinction [3]. Indeed, mounting evidence attests to the rapid reconfiguration of species distributions along depth or latitudinal gradients in response to changing climates [4,5]. Marine mammals, serving as highly susceptible apex predators within marine ecosystems, assume a pivotal role in maintaining biodiversity and regulating ecosystem processes [6]. Consequently, obtaining comprehensive and precise knowledge of the current distributions of marine mammals, as well as reliable projections under future climate change scenarios, is of paramount importance in developing effective strategies for resource management and conservation.

Species distribution models (SDMs) are useful tools in this endeavor, as they can predict potential changes in species habitats by determining statistical relationships between species occurrence data and environmental predictors, and can also be used to forecast how suitable areas may vary under different climate change scenarios [7]. Traditionally, SDMs have been constructed at the species level based on the “niche conservatism” hypothesis, which suggests that individuals of the same species have similar niche spaces and exhibit consistent responses to climate change across their range [8,9]. Nevertheless, over an extensive evolutionary timeframe, species with a wide range may gradually adapt to local conditions, leading to niche divergence [10]. Recent research highlights a growing recognition of the importance of integrating local adaptation or intraspecific variation into climate responses, with an increasing number of studies emphasizing this crucial aspect [11,12]. By modeling habitat suitability below the species level, more accurate estimations of species ranges and climate change projections can be attained. Notably, within the realm of species conservation, the consideration of climate change responses within phylogeographic lineages has gained currency for certain taxonomic groups in terrestrial communities [13,14], and marine communities [15,16].

Here, we examined how predictions of climate change responses can differ when considering local adaptation for spotted seals. The International Union for Conservation of Nature’s Red List of Threatened Species ranked spotted seals (*Phoca largha*) as “least concern”, indicating a low risk of extinction [17]. However, in China, spotted seals have the highest protection level among rare and endangered species (class-I protection) due to increasing threats from climate change and habitat loss [18]. North Pacific spotted seals have eight specialized breeding habitats and limited mobility, which contribute to their vulnerability to climate change [19]. Based on morphological and genetic differences, spotted seals were divided into three distinct geographical populations [20]: the Bering population (BDPS), the Okhotsk population (ODPS) and the Southern population (SDPS). Throughout their extensive evolutionary narratives, the three populations have taken up residence within divergent ecological domains, thereby hinting at the plausible presence of localized adaptation. While the precise degree of distributional overlap and genetic interchange amongst these populations remains shrouded in obscurity, the likelihood of their existence cannot be discounted [21,22]. Neglecting to account for such local adaptations when employing SDMs in conservation or management decisions may result in erroneous characterizations of species’ responses to environmental changes throughout their ranges, thus misdirecting conservation efforts [23]. Therefore, to further estimate the impacts of climate change on spotted seals at a higher taxonomic resolution below the species level has become an urgent issue in their long-term conservation and management.

In this study, we quantified realized niches (i.e., the portion of the fundamental niche currently used by the species), developed SDMs and made future predictions to examine how climate change might influence spotted seals by constructing species-level versus population-level models. We sought to address the following hypotheses: (1) conspicuous disparities exist in spatial niches among the three distinct populations; (2) the three populations respond differently to climate change, and population-level SDMs are more reliable than species-level results; (3) the potential distribution of spotted seals under future climatic conditions will shift to higher latitudes; and (4) there are large gaps in spotted seal conservation outside protected areas that are not effectively protected. Our results emphasize the value of developing SDMs below the species level and serve as a useful guide for designing climate-adapted conservation and management strategies for spotted seals within more precise taxonomic units.

## 2. Materials and Methods

### 2.1. Study Area and Spotted Seal Occurrence Data

This study focused on the distribution range of spotted seals in the North Pacific, with the main study area located in the temperate and cold temperate coastal and littoral regions (90° E—240° W, 0° N—80° N; Figure 1). The spotted seal occurrence data were collected from the literature, the Global Biodiversity Information Facility (GBIF; http://www.gbif.org/, accessed on 21 December 2022) [24], and the Ocean Biogeographic Information System (https://obis.org/, accessed on 4 December 2022) [25] (Appendix A). To minimize sampling bias, we spatially thinned the occurrence data to match the resolution of the environmental data (5′ × 5′, approximately 9.2 km × 9.2 km) using the R package “spThin” [26], and only one random spotted seal distribution record was used in each raster. Following this data-filtering procedure, 1990 records were kept in order to construct the SDM at the species level (hereafter “species model”). Of these, 680 records belonged to the BDPS that was used to construct the SDM at the population level (hereafter “BDPS model”), 853 records belonged to the ODPS that was used to construct the ODPS model, and 457 records belonged to the SDPS that was used to construct the SDPS model.

### 2.2. Environmental Predictor Variables

Habitat surroundings have a significant influence on the distribution of spotted seals, and considering a combination of bioenvironmental relevance and data availability, 10 environmental variables that may influence the distribution of spotted seals were selected for this study (Table 1). Current and future environmental data were downloaded from online datasets: the water depth and distance to shore were downloaded from the Global Marine Environment Datasets (https://gmed.auckland.ac.nz/, accessed on 15 April 2023) [27], and the remaining predictors were downloaded from the Ocean Raster for Analysis of Climate and Environment (https://bio-oracle.org/, accessed on 6 April 2023) [28]. Considering the correlation between predictors, we completed Pearson’s correlation factor analysis between environmental layers using the R package usdm [29], retaining only environmental variables with correlation factor values < |0.7| [30] (Appendix A). Finally, seven predictors, including water depth (Dep), distance to shore (DTS), chlorophyll concentration (Chl), current velocity (CV), ice thickness (IT), salinity, and water temperature (Tmean), were retained for modeling analysis.

To project the future habitat suitability of spotted seals, we considered four representative concentration pathway (RCP) scenarios (i.e., RCP 2.6, RCP 4.5, RCP 6.0 and RCP 8.5), and two periods (i.e., 2050s: the average for 2040s–2050s, and 2100s: the average for 2090–2100). We obtained the corresponding projections of future marine environmental layers from Bio-ORACLE v2.0. This study assumed no change in water depth and distance to shore [31].

### 2.3. Estimates of Niche Divergence

To assess whether the three spotted seal populations occupy different niche spaces, we used n-dimensional hypervolume to characterize their realized niche [32]. For this, we first performed principal component analysis (PCA) on seven selected environmental variables and retained the top four principal components, which cumulatively explained 83.6% of the total variance (Appendix A). Then, we calculated the principal component retention values corresponding to each population using a Gaussian approach with the R package hypervolume [33]. Finally, the niche divergence between populations can be assessed by overlapping the hypervolume of each population using the R package BAT [34]. Total niche divergence (βTotal) was divided into the following two processes: niche shift (spatial replacement between hypervolumes) and niche contraction/expansion (net difference between hypervolumes). The βTotal ranged from 0 to 1, indicating the two hypervolumes of complete overlap to complete separation [11].

### 2.4. SDMs Establishment and Projection

We conducted SDM analysis based on the “biomod 2” package in the R platform (version 4.2.3) [35]. This package contained 10 modeling algorithms: generalized additive mode, generalized linear model (GLM), generalized boosting model (GBM), random fores, surface range envelope (SRE), artificial neural network (ANN), flexible discriminant analysis (FDA), classification tree analysis (CTA), multiple adaptive regression splines (MARS), and maximum entropy (Maxent). Since true absence data were lacking, we randomly simulated the same number of pseudo-absence records as that of presence records in the environmental conditions [36]. The dataset was divided into five groups during the modeling process, with an equal number of records in each group, four of which were used for model training and the remaining one for model testing. To evaluate the predictive performance of each model, the 5-fold cross-validation process was repeated 10 times. We used the TSS (the true skill statistic) and AUC (the area under the ROC curve) values to assess the accuracy of the models.

This study selected TSS > 0.8 and AUC > 0.9 as model selection standards [37,38] and used a weighted-average algorithm to build an integrated model for reducing the uncertainty of individual models. To better explain habitat suitability, we transformed continuous habitat suitability predictions into a binary map by maximizing the probability threshold of the TSS [39]. We applied a randomized method to measure Pearson correlations between all predictor and assessment variables [40] to assess the relative importance of each variable in predicting species distributions. Finally, we built two levels of species and population ensemble models to predict the potential distribution of habitat for the whole species and three geographical populations (the BDPS, ODPS and SDPS) under current and future (2050s, 2100s) climate scenarios under RCP 2.6 and RCP 8.5.

### 2.5. Protection Gap Analysis

The Global Marine Protected Areas layers were sourced from the World Database on Protected Areas (https://www.protectedplanet.net/, accessed on 4 May 2023) [41], while data on protected areas in the Yellow and Bohai Seas of China were sourced from a published article [42]. First, we overlaid the layers of existing protected areas and the range of spotted seals to analyze the proportion of the existing protected area covered in the spotted seal distribution range and the uncovered spatial area in QGIS 3.28.6 software (https://www.qgis.org/en/site/, accessed on 5 April 2023). We then conducted a conservation gap analysis of the uncovered spatial areas to identify uncovered habitat areas. Finally, by integrating current and future climate change scenarios, we projected trends in the range of spotted seals and further identified conservation gaps for these species under future climate scenarios.

## 3. Results

### 3.1. Niche Divergence among the Three Populations

According to the results of niche divergence studies of different geographical populations, the BDPS has the widest ecological range. The four-dimensional hypervolume for the BDPS, ODPS, and SDPS, respectively, was 1642.20, 353.46, and 145.16. The niche divergence between two populations was very high, with values of 0.81 (BDPS:ODPS), 0.92 (BDPS:SDPS), and 0.86 (ODPS:SDPS) shown by the paired comparison of hypervolumes. Contraction/expansion accounted for more than 85% of the niche divergence between the BDPS and the ODPS or SDPS, whereas niche transitions had a much smaller role (15%). The main cause of the niche divergence between the ODPS and SDPS was contraction/expansion (>65%), with niche shift accounting for a little part (35%) (Table 2).

When the three populations’ general niches were compared, it was clear that PCA1 was the primary focus of niche divergence (Figure 2a), which was mostly explained by water depth, distance to shore, and chlorophyll concentration (Appendix A). The paired-niche comparison revealed that the PCA1 was primarily responsible for the niche divergence between the BDPS and ODPS (Figure 2b), which was primarily explained by water depth, ice thickness, and current velocity (Appendix A); the PCA1 was responsible for the niche divergence between the BDPS and SDPS (Figure 2c), which was primarily explained by water temperature, chlorophyll concentration, and salinity (Appendix A); and the PCA1 was also responsible for the niche divergence between the BDPS and SDPS (Figure 2d), which was primarily explained by ice thickness, chlorophyll concentration and water temperature (Appendix A).

### 3.2. Current SDMs Projections

Based on the TSS and AUC values in the model results, eight models were selected to build the weighted species-level ensemble model after removing MaxEnt and SRE, and nine models were selected to build the weighted population-level ensemble model after removing SRE from the ten single models. The higher values of AUC and TSS for all four ensemble models indicated high predictive performance (Table 3).

The species-level model showed that water temperature and depth considerably contributed to the distribution of spotted seals, while current velocity and chlorophyll concentration contributed little to the model (Figure 3a). According to this model, spotted seals preferred to live in waters with a temperature range of 0 °C to 15 °C and a depth of 0 m to 1000 m (Appendix A). The population-level model showed that the main factors influencing the potential distribution of spotted seals differ between populations. Specifically, the BDPS distribution was most influenced by water temperature and ice thickness; this population preferred to inhabit areas with water temperature ranging from 0 °C to 8 °C (Appendix A) and had a higher probability of occurrence at ice thicknesses of 0–1 m (Appendix A). The ODPS distribution was most influenced by water temperature and depth; this population preferred to inhabit areas with water temperature ranging from 2 °C to 12 °C (Appendix A) and water depth ranging from 0 m to 2000 m (Appendix A). The SDPS distribution was mainly influenced by water temperature and distance to shore (Figure 3b); this population preferred to live in a water temperature range of 2 °C to 15 °C (Appendix A) and had a higher probability of living in near-shore waters not more than 800 meters from the coast (Appendix A).

According to the results of the model, the distribution and suitable habitat of the three spotted seal populations under current environmental conditions are as follows: the BDPS is mainly distributed in the Chukchi Sea, the Bering Sea, the coast of Kamchatka Island and the northern part of the Sea of Okhotsk, with the largest area of suitable habitat (6.24 × 10^6^ km^2^) (Figure 4c,f). The ODPS is mainly distributed in the Sea of Okhotsk and near Sakhalin Island, extending northward to the Aleutian Peninsula, with the second largest area of suitable habitat (3.54 × 10^6^ km^2^) (Figure 4d,g). The SDPS is mainly distributed on Hokkaido Island, the Sea of Japan and the Yellow Sea of China, reaching as far as the East China Sea, with the smallest area of suitable habitat (1.08 × 10^6^ km^2^) (Figure 4e,h).

The species model results are shown in Figure 4a,b. Both the continuous and binary projections showed that the predicted suitable area from the species model is similar to that from the BDPS model in the Chukchi Sea and the Bering Sea, is similar to that from the ODPS model in the Okhotsk Sea and is smaller than that from the SDPS model in the southernmost region.

### 3.3. Habitat Suitability under Future Climate Scenarios

This study built SDMs at both the species and population levels to predict climate change impacts on potentially suitable habitats for spotted seals. Habitat-scale changes are influenced by climate change scenarios, particularly under the pessimistic scenario of uncontrolled greenhouse gas emissions (RCP 8.5), and the suitable habitat is predicted to vary considerably within the appropriate range (Table 4). For all climate change scenarios, both the BDPS and ODPS show a decreasing trend in the extent of suitable habitat, with the greatest decrease under RCP 8.5 in the 2050s, while the SDPS shows an increasing trend in the extent of suitable habitat. For the BDPS, suitable habitat for spotted seals is mainly stable in the Chukchi Sea and around the Bering Strait, with the northern coasts of the Chukchi and Taymyr Peninsulas also serving as potential habitats (Figure 5c,f). For the ODPS, suitable habitat is mainly stable along the northwestern coast of the Kamchatka Peninsula, while most of the suitable habitat in the Sea of Okhotsk will be lost (Figure 5d,g). For the SDPS, although large areas of suitable habitat are found around the Aleutian Islands and the Gulf of Alaska, the southernmost spotted seal colony in Liaodong Bay will be lost (Figure 5e,h).

At the species level, the southernmost Liaodong Bay spotted seal breeding area is gradually being lost; as the range of the Arctic Ocean seas north of the Chukchi Peninsula expands toward higher latitudes, the distribution of spotted seals may occur at higher latitudes in the Thamel Peninsula (Figure 5a,b). Range changes in the size of the predicted species tended to decrease under RCP 2.6 and increase under RCP 8.5, with the change reaching 36.16% under RCP 8.5 in the 2100s.

### 3.4. Spotted Seal Conservation Gap Analysis

According to the results of the protection gap analysis, 278,617 km^2^ was found to be protected, representing only 5.65% of the spotted seal range, i.e., more than 94% of the area is not covered by protected areas (Figure 6a). In the overlaid map, the Eastern Bering Sea, the Aleutian Islands and the Sea of Okhotsk overlap significantly with the range of spotted seals. Although the number of protected areas around the Yellow and Bohai Seas in China is high, they are very small. Further analysis suggests that outside these existing marine protected areas (MPAs), there are both important nonbreeding areas and breeding habitat for spotted seals, such as the western coast of the Kamchatka Peninsula, the western Bering Sea and Peter the Great Bay, which remain unprotected.

In this study, the potential range of spotted seals under future climate scenarios was overlaid with existing MPAs, and it was found that spotted seals were protected in only 6.04–6.22% of their range under the RCP 2.6 and RCP 8.5 scenarios in the 2050s and 2100s (Table 5). In summary, the overall change in the protected gap area is insignificant in the four future climate scenarios (Figure 6b,c).

## 4. Discussion

### 4.1. Consideration of Local Adaptation

The study quantified the realized niches of three spotted seal populations and found that the BDPS exhibited a substantially broader niche space than the ODPS and SDPS, primarily due to niche contraction/expansion. These findings underscore that geographically separated spotted seal populations inhabit dissimilar ecological niches; therefore, niche conservatism does not apply to this species. Then, we constructed SDMs for the BDPS, ODPS, and SDPS individually, which accounted for local adaptation, and these models revealed that the three populations showed differing responses to climate change predictors. In summary, this study emphasizes the importance of considering local adaptation in projecting the potential distribution of species to inform conservation and management decisions in a climate change scenario.

Local adaptation and intraspecific variation were incorporated into SDMs based on the recognition that populations of species inhabiting widely different habitats over significant time scales will often show adaptations to their respective local conditions, resulting in intraspecific niche variation. This local adaptation may be reflected by morphological and genetic differences [43]. For instance, studies have revealed significant differences in the nonmetric features of the skull between spotted seals from the central and eastern Bering Sea [44] and a phylogeographic break between spotted seals breeding in the Yellow Sea–Japan Sea region and those breeding in the Sea of Okhotsk, Bering Sea and Chukchi Sea. The above studies, in conjunction with the observed niche divergence among the BDPS, ODPS, and SDPS established in this study, stress the importance of building SDMs at the population level to account for local adaptation.

In this study, despite the fact that the species and population level SDMs predicted a similar change trend in the species range, with a reduction in suitable habitat for the BDPS and ODPS and an expansion of suitable habitat for the SDPS under future climate change scenarios, the magnitudes of the range change predicted by the two types of models varied. The population level model produced more encouraging findings for the BDPS and ODPS with less loss of appropriate habitat. Due to the inclusion of potential local responses in population-level models, our climate change estimates were in fact less pessimistic. Our results are in line with a number of published studies that suggest adaptive genetic variation within a species can reduce the species’ susceptibility to climate change [45,46,47].

### 4.2. Impacts of Climate Change on Spotted Seals

The niche divergence of the BDPS, ODPS and SDPS was mainly due to niche contraction/expansion, to which water temperature, ice thickness and chlorophyll concentration contributed most. Spotted seals are cool-temperature marine mammals and water temperature is an important influence on both the physiology and behavior of spotted seals, affecting them indirectly by altering the distribution of prey, predators and disease-causing vectors [48]. Also, spotted seals are often dependent on sea ice for breeding and foraging, and thicker ice may provide better breeding and foraging conditions, while thinner ice may limit these activities. Moreover, chlorophyll concentration variations may reflect the productivity of marine ecosystems and the base of the food chain [49]. Differences in nutrient conditions and chlorophyll concentration in different regions may lead to different availability of food resources, and spotted seal populations may choose to adapt to different food resources according to chlorophyll concentration in different regions, resulting in population differentiation. Climate-induced changes in these factors cause species to alter their current distribution patterns to track ecological niches. In summary, these three climatic factors directly or indirectly affect the physiology, behavior and availability of food resources of spotted seals, thus leading to the divergence of adaptation strategies among different populations.

Different populations of spotted seals may have different levels of adaptiveness and vulnerability to climate change. Our population-level SDMs predicted that the BDPS will colonize the northern coast of the Chukchi, and as far west as the Taymyr Peninsulas, while retaining most of the current suitable areas, indicating the resilience of the population to climate change. In contrast, the ODPS preserves most of the currently suitable areas, and parts of the suitable habitat in the Sea of Okhotsk will be lost, indicating that the population is less resilient to climate change. The SDPS will have large areas of suitable habitat near the Aleutian Islands and the Gulf of Alaska, but spotted seals of the SDPS would not be capable of shifting their range to the northeast to reach this area and the southernmost spotted seal breeding area in Liaodong Bay will be lost, indicating the vulnerability of this population to climate change.

Overall, the thinning and breaking up of ice caused by climate change will expose vast regions of the northern Bering Sea and Chukchi Sea, which is likely to increase suitable habitat for northern spotted seals, thus offsetting the loss of habitat in the south. Similar findings were reported in studies of climate change-induced geographic translocation of species [15,50,51,52].

### 4.3. Model Predictive Accuracy

Integrated habitat suitability models built with a weighted integration technique, especially for rare species, can increase model prediction accuracy and avoid overfitting problems without compromising explanatory power [53]. The TSS values of the four ensemble models were all 0.86, their AUC values were all 0.95, and their errors for the environmental importance results were all less than 0.2, showing that the predictions were very accurate representations of the current and future distributions of spotted seals under various climate scenarios. An integrated model was used in this work to estimate the whole distributions of spotted seals, and the results were mostly in line with the known range of spotted seals (temperate and cold temperate coastal and shore).

Although the results were as expected, there were limitations to the method of model species distribution to predict the range of spotted seals. We used future environmental data that did not occur objectively but were predictions based on atmosphere-ocean general circulation models (AOGCMs), as well as the effects of many factors such as changes in food availability, changes in ocean currents that persist between breeding sites, and the ability of spotted seals to swim on their own. When predicting the suitable distribution of spotted seals, the results of the study may have overestimated the range of the spotted seal. Predictions made in the future that take these elements into account in the model will be closer to the species’ actual distribution. Moreover, to improve or validate the predictive power of SDMs, independent geographically or temporally separated data should be collected [54]. Considering the difficulty and cost of field surveys, emerging environmental DNA methods could be used to determine the presence of potential distribution areas for spotted seals as predicted in this study, and these data could be used in future work on SDMs.

### 4.4. Management and Conservation of Spotted Seals

Spotted seals have received little attention from conservationists or managers despite significant human and climate change challenges. Spotted seals and other marine mammals are vulnerable in the face of global warming, and their potential extinction could have far-reaching consequences for the functioning of global marine ecosystems in the future [55]. Water temperature and ice thickness are two important environmental factors that might affect the distribution of spotted seal populations geographically, yet they are also strongly related to climate change. To develop climate-adapted conservation and management methods, it is imperative to assess how the changing climate is affecting the appropriateness of spotted seal habitat. In general, all three populations exhibit a propensity to migrate toward the poles under a warming climate. However, the SDPS are more vulnerable to climate than the BDPS and ODPS, and their population sizes have already been significantly reduced from historical levels and may be at risk of population genetic extinction. Therefore, the SDPS deserves more attention and protection in the face of climate stress.

Marine protected areas (MPAs) have proven to be an effective tool for protecting endangered species and maintaining ecosystem services [56]. The European Union target of ‘30 by 30’ that is 30% of the ocean protected (as MPAs) by 2030. For the conservation of spotted seals and their habitat, many countries and regions have established MPAs [57,58]. These marine protected areas were crucial for the conservation of spotted seals and their habitats. However, when we overlaid the spotted seal range, we discovered that only 5.65–6.22% of the range was protected, meaning that more than 94% of the area was unprotected. Therefore, protected areas (marine reserves, nature reserves and national parks) need to be expanded, and the establishment of protected areas across international borders should be considered to better protect spotted seals and their habitats [59].

The current and potential future distribution of spotted seals is mainly in coastal waters; as a result, protecting the species from anthropogenic environmental contamination and hunting pressure in these seas is essential. Given the different vulnerabilities of the three populations to climate change, we need to develop local adaptive conservation and management measures for the populations in different areas. Since the SDPS is most sensitive to climate change, we recommend that stricter hunting restrictions, such as bans on poaching for genitalia and culling by fishers, be imposed on this population. For ODPS, although there is a degree of adaptability to climate change and little risk of population extinction, direct or indirect commercial fisheries interactions may have a significant cumulative effect. Therefore, we should increase management efforts to control marine development activities such as sand mining, oil and gas exploration and water pipelines to reduce damage to spotted seals (survival environment). Although the BDPS has the ability to adapt to climate change, we should also pay attention to the timing and routes of the breeding migrations of spotted seals, avoid fishing and shipping operations during migrations, and raise awareness of conservation.

Finally, we must emphasize that MPAs and other suggested conservation measures will only guarantee that adequate habitats for spotted seals are safeguarded from human impact now and in the future. However, as greenhouse gas concentrations rise, the SDPS will lose its ideal habitat. This population will progressively decline towards extinction if the issue is not managed for the long-term. Therefore, reducing human-caused greenhouse gas emissions is the ultimate solution for the sustainability of these populations.

## 5. Conclusions

In conclusion, this study represents the first step in estimating climate impacts on the potential distribution of spotted seals in the North Pacific considering local adaptation. Population-level SDMs are more reliable than species-level SDMs because of the different responses of the three spotted seal populations to environmental predictor variables. Additionally, conservation efforts should be dedicated to the establishment of MPAs, first in the stable spots predicted to remain climatically suitable for the species, and second in the currently suitable areas. The comparison of current and predicted habitat suitability maps presented in our study serves as a crucial tool allowing us to delineate the most promising regions for establishing both types of measures. In future studies, other analytical methods and multiple data sources should be incorporated to improve our ability to predict the potential distribution of spotted seals and deliver more accurate information for related conservation and management.

## Figures and Tables

**Figure 1 animals-13-03260-f001:**
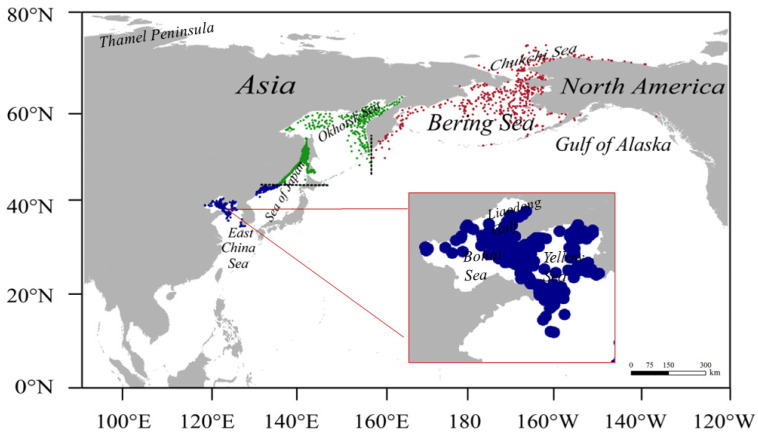
Map of the study area and occurrence records of spotted seal. Red dots represent the occurrence records of the Bering population (BDPS), green dots represent those of the Okhotsk population (ODPS), and blue dots those of the southern population (SDPS). The dotted black lines are drawn along 43° N latitude and 157° E longitude, which are considered the boundaries between the SDPS and ODPS and between the ODPS and BDPS.

**Figure 2 animals-13-03260-f002:**
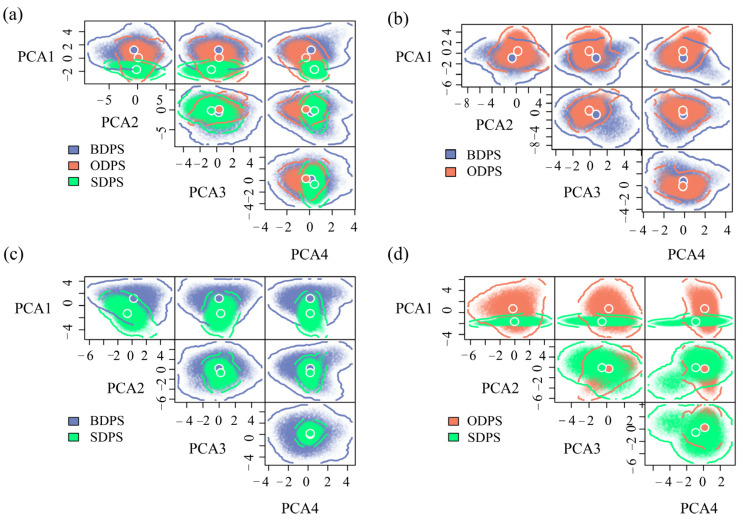
Three spotted seal populations with realized niches quantified by four-dimensional hypervolume. The larger blue, orange and green dots indicate the mean niche position (niche centroid) of the Bering population (BDPS), Okhotsk population (ODPS), and southern population (SDPS), respectively. The overlapping hypervolumes for the three populations (**a**), BDPS and ODPS (**b**), BDPS and SDPS (**c**) and ODPS and SDPS (**d**).

**Figure 3 animals-13-03260-f003:**
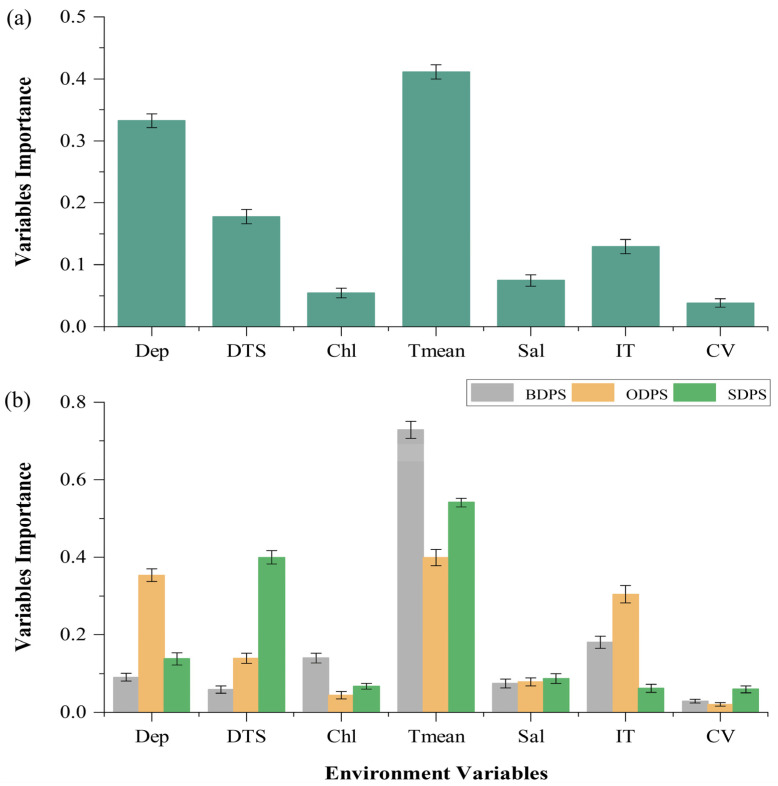
Importance of environmental variables in driving whole-species distribution based on the species-level model (**a**) and population distributions based on the population-level model (**b**). Dep, water depth; DTS, distance to shore; Tmean, water temperature; Sal, salinity; IT, ice thickness; CV, current velocity.

**Figure 4 animals-13-03260-f004:**
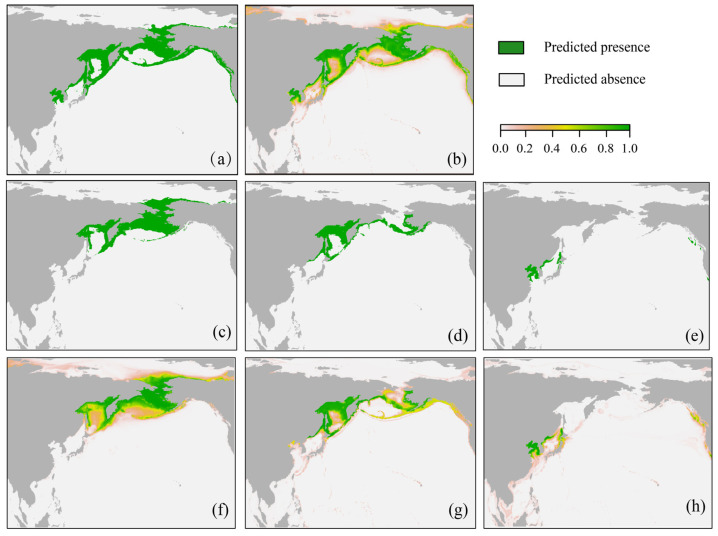
Habitat suitability map of spotted seals projected by the ensemble models under current climate scenarios. (**a**,**b**) Respective binary and continuous plots of the species; (**c**,**f**) respective binary and continuous plots for BDPS; (**d**,**g**) corresponding plots for ODPS; (**e**,**h**) corresponding plots for SDPS.

**Figure 5 animals-13-03260-f005:**
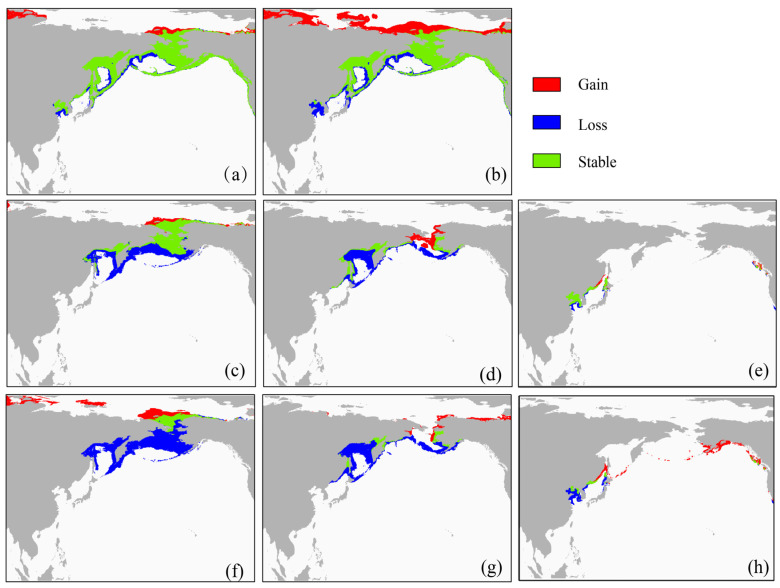
Predicted change in the suitable habitat based on the ensemble models in the 2050s under RCP 8.5 (5085) and in the 2100s under RCP 8.5 (0085) for the species (**a**,**b**), BDPS (**c**,**f**), ODPS (**d**,**g**) and SDPS (**e**,**h**).

**Figure 6 animals-13-03260-f006:**
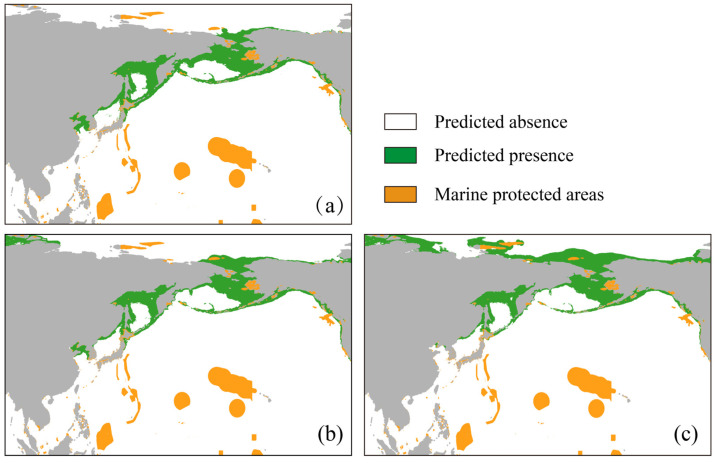
Analysis of the conservation gap for spotted seals under current and future climate scenarios. (**a**) Analysis of the conservation gap under the climate scenario; (**b**) analysis of the conservation gap in the 2050s under RCP 8.5 (5085); (**c**) analysis of the conservation gap in the 2100s under RCP 8.5 (0085).

**Table 1 animals-13-03260-t001:** Ten environmental variables initially selected for this study and their units, spatial resolution and sources.

Environment Variable	Unit	Spatial Resolution	Source
water depth	m	5 arc minutes	https://gmed.aucklandac.nz/, accessed on 15 April 2023
distance to shore	km	5 arc minutes	https://gmed.auckland.ac.nz/, accessed on 15 April 2023
calcite	mol·m^−3^	5 arc minutes	https://bio-oracle.org/, accessed on 6 April 2023
chlorophyll concentration	mg·m^−3^	5 arc minutes	https://bio-oracle.org/, accessed on 6 April 2023
currents velocity	m·s^−1^	5 arc minutes	https://bio-oracle.org/, accessed on 6 April 2023
dissolved oxygen	mol·m^−3^	5 arc minutes	https://bio-oracle.org/, accessed on 6 April 2023
sea ice concentration	fraction	5 arc minutes	https://bio-oracle.org/, accessed on 6 April 2023
ice thickness	m	5 arc minutes	https://bio-oracle.org/, accessed on 6 April 2023
salinity	PSS	5 arc minutes	https://bio-oracle.org/, accessed on 6 April 2023
water temperature	°C	5 arc minutes	https://bio-oracle.org/, accessed on 6 April 2023

**Table 2 animals-13-03260-t002:** Total niche differentiation (βTotal) between populations and the proportion of niche shift and niche contraction/expansion. BDPS, Bering distinct population segment; ODPS, Okhotsk distinct population segment; SDPS, Southern distinct population segment.

Populations Pair	βTotal	Niche Shift	Niche Contraction/ Expansion
BDPS-ODPS	0.81	0.12(15%)	0.69(85%)
BDPS-SDPS	0.92	0.04(4%)	0.88(96%)
ODPS-SDPS	0.86	0.27(32%)	0.59(68%)

**Table 3 animals-13-03260-t003:** Number of models and evaluating indicators for the ensemble models built at the species level and population level. NME, number of models used in ensemble modeling; TSS, the true skill statistics; AUC, the area under the receiver operating characteristic curve.

Ensemble	TSS	AUC	NME
Species model	0.857	0.953	8
BDPS model	0.861	0.949	9
ODPS model	0.932	0.975	9
SDPS model	0.950	0.978	9

**Table 4 animals-13-03260-t004:** Predicted size of changes [9] in species range based on the species-level and population-level models under future climate scenarios. RCP 2.6 (8.5), the representative concentration pathway 2.6 (8.5); 2050s (2100s), at the middle (end) of the 21st century.

RCP	BDPS		ODPS		SDPS		Species	
	2050s	2100s	2050s	2100s	2050s	2100s	2050s	2100s
RCP 2.6	−32.48	−38.34	−32.80	−47.61	13.15	17.86	−1.43	−1.73
RCP 8.5	−38.24	−63.94	−44.25	−66.51	9.57	62.91	2.27	36.16

**Table 5 animals-13-03260-t005:** Spotted seal areas (km^2^) and corresponding percentages protected under the current and future climate scenarios.

Climate Scenario	Area Protected (km^2^)	Percentage of Protection [9]
current	278,617	5.65
2050s RCP 2.6	278,062	6.17
2050s RCP 8.5	277,776	6.04
2100s RCP 2.6	279,403	6.22
2100s RCP 8.5	277,449	5.56

## Data Availability

The data that support the findings of this study are available from the corresponding author upon reasonable request.

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
