# Peer review of "Estimating the Spatial Distribution and Future Conservation Requirements of the Spotted Seal in the North Pacific"

_animals, 2023, doi:10.3390/ani13203260_

Round 1

Reviewer 1 Report

2.1    Study area and spotted seal occurrence data

Please add the time for accessing the database.

Line113, please change spThin to "spThin".

2.2    Environmental predictor variables

  Please add the time for accessing the database.

2.3    Protection gap analysis

Line191, please add the time for accessing the database.

Please add the website to obtain the software after QGIS3.28.6

2.4    SDMs establishment and projection

Line166, please change biomod 2 to " biomod 2".

3.2 Current SDMs projections

(1) How do these key environmental variables specifically affect the potential distribution of these organisms? For example, how does their probability of existence change with variations in water temperature? how does their probability of existence change with increasing water depth? Furthermore, it is important to provide a thorough and detailed explanation of these findings in your discussion.

(2) Is it more appropriate to use scientific notation when analyzing the potential distribution area of a species.

4.4 Management and conservation of spotted seals

(1) You mentioned the impacts of anthropogenic environmental pollution and hunting pressure on species here, and provided a series of management suggestions. However, I regret that I did not see specific conservation recommendations and management strategies from you regarding the key environmental variables that affect the potential distribution of species' geographic populations.

The manuscript needs language refinement before it can be published

Author Response

Dear reviewer,

Thanks very much for your constructive commetns and suggestions to our manuscript. We have carefully revised our paper according to your comments. The responses to your commnets is attached.

Reviewer 2 Report

Overall, a well conceived and written manuscript.

Below I give both specific suggestions and corrections (by line number) and some  general points that I think could be addressed by the authors and inserted where the authors (or Editor) feel most appropriate.

I suggest the title should read something like: Estimating the spatial distribution and future conservation requirements of the spotted seal.....

line 25: replace 'estimations' with 'requirements'

line27-28: replace 'different regions' with 'its range'

line 32: replace 'There was' with 'We predicted'

line 34: replace 'neglected by' with 'not represented in'

line 57: replace 'potent' with 'useful'

line 76: replace 'meaning' with 'indicating'

line 89: delete 'how'

Figure 1 (and line 125): the population boundary between ODPS and SDPS at 43oN seems rather arbitrary splitting an apparently continuous range in the Sea of Japan. Is there a reference that can be cited to justify this?  Zhuang et al. (2023) make the point that it is the Yellow Sea population that is isolated and genetically distinct.

line 140: typo 'salinity'

line 192: some geographical locations, e.g. Bohai Sea (and Liaodong Bay in the abstract) are not marked on Figure 1. Could an second map be inserted to show these locations within the wider Yellow Sea area? 

line 307: where is the Thamel Peninsula? If it would be difficult to place on Figure 1 perhaps coordinates could be given here?

line 317: 'areas of survival' is rather clumsy English. Perhaps rephrase this sentence 'there are both important non-breeding areas and breeding habitat for.....

line 310: suggest delete 'are often uncovered and therefore'

line 346: maybe insert 'in a climate change scenario' at the end of the sentence

line 356: replace 'imperative' with 'importance'

line 377: rejig sentence to read 'northern coast of the Chukchi, and as far west as the Taymyr, Peninsulas,  while retaining'

line 381: perhaps insert a sentence as to whether spotted seals of the SDPS would be capable of shifting their range to the northeast reach this area?

line 421: relocate comma

Section 4.4: In this first paragraph a couple of sentences could be drafted to introduce ideas such as the European Union target of '30 by 30' that is 30% of the ocean protected (as MPAs) by 2030.

line 434: perhaps place 'survival environment' in parentheses

line 441: suggest 'This population will progressively decline towards extinction if the issue.....'

References: please place all Latin species names in italics

All improvements to English are given above

Author Response

(The authors gave the same response as above.)
